# Assessing G4-Binding Ligands In Vitro and in Cellulo Using Dimeric Carbocyanine Dye Displacement Assay

**DOI:** 10.3390/molecules26051400

**Published:** 2021-03-05

**Authors:** Nakshi Desai, Viraj Shah, Bhaskar Datta

**Affiliations:** 1Department of Biological Engineering, Indian Institute of Technology, Gandhinagar 382355, India; nakshi.desai@iitgn.ac.in (N.D.); viraj.s@iitgn.ac.in (V.S.); 2Department of Chemistry, Indian Institute of Technology, Gandhinagar, Gandhinagar 382355, India

**Keywords:** DNA G-quadruplex, RNA G-quadruplex, G4 ligands, fluorescence displacement assay, G4 visualization, polymerase stop assay

## Abstract

G-quadruplexes (G4) are the most actively studied non-canonical secondary structures formed by contiguous repeats of guanines in DNA or RNA strands. Small molecule mediated targeting of G-quadruplexes has emerged as an attractive tool for visualization and stabilization of these structures inside the cell. Limited number of DNA and RNA G4-selective assays have been reported for primary ligand screening. A combination of fluorescence spectroscopy, AFM, CD, PAGE, and confocal microscopy have been used to assess a dimeric carbocyanine dye B6,5 for screening G4-binding ligands in vitro and in cellulo. The dye B6,5 interacts with physiologically relevant DNA and RNA G4 structures, resulting in fluorescence enhancement of the molecule as an in vitro readout for G4 selectivity. Interaction of the dye with G4 is accompanied by quadruplex stabilization that extends its use in primary screening of G4 specific ligands. The molecule is cell permeable and enables visualization of quadruplex dominated cellular regions of nucleoli using confocal microscopy. The dye is displaced by quarfloxin in live cells. The dye B6,5 shows remarkable duplex to quadruplex selectivity in vitro along with ligand-like stabilization of DNA G4 structures. Cell permeability and response to RNA G4 structures project the dye with interesting theranostic potential. Our results validate that B6,5 can serve the dual purpose of visualization of DNA and RNA G4 structures and screening of G4 specific ligands, and adds to the limited number of probes with such potential.

## 1. Introduction

G-quadruplexes (G4s) are Hoogsteen hydrogen bond stabilized nucleic acid secondary structures present in several regulatory regions of the human genome, such as telomeres and oncogene promoters [1,2,3]. These structures have been the focus of intense research owing to their implication in regulation of gene expression, RNA metabolism, epigenetic remodeling, and the prognosis of several diseases [3,4]. Research into the physiological relevance of G4s has necessitated methods by which these structures can be studied. The in vitro formation of G4 structures has been studied by employing a variety of techniques including NMR, circular dichroism (CD), X-ray crystallography, UV-visible, and fluorescence spectroscopy [5]. The in cellulo study of G4s is significantly more challenging owing to the complex polymorphic and transient character of the structures [6]. Chemical biology-based strategies leveraging molecules that are capable of selectively binding G4s, enable scrutiny of structure-function correlations of the secondary structures. In this regard, a limited number of G4 selective chromophores have been reported that exhibit multi-fold fluorescence enhancement upon encountering G-quadruplexes. These agents include cyanine, carbazole and pyridinium derivatives, and are capable of distinguishing quadruplex structures from canonical duplex DNA [7,8,9]. One approach towards conception of such probes is to associate fluorophores with exhibit G4-selective ligands in a modular manner. This approach suffers from the drawback of stabilization of G4 structure at the expense of fluorescence quantum yield [10]. Another approach involves fusing molecules that stabilize G-quadruplexes with a fluorescent agent. Molecules in this category include Thioflavin T, TMPyP4, porphyrins, and phthalocyanines [11]. Such molecules have been used for the visualization of G4 structures and evaluation of G4-targeting drugs by fluorescence displacement assay. Several assays have been reported for the efficient discovery and characterization of G4 ligands. These include chip-based analysis [12,13], ligand fishing [14], Förster resonance energy transfer (FRET) melting [15], and equilibrium dialysis [16]. Speed and cost of analysis are major constraints for robust development of these methods. Thiazole orange (TO) displacement assay is a widely reported screening tool of G4 binding molecules with fluorescence quenching as readout [17]. Alternatively, PhenDV-based G4- fluorescence intercalator displacement assay is a way to screen medium and high affinity G4 ligands, where readout is by fluorescence enhancement and not quenching [18]. However, a very high affinity of PhenDV for G4 structures in c-myc makes it incapable to identify many effective G4-targeting molecules to be analyzed through this displacement assay [19]. Only a limited number of RNA G4-specific probes have been reported along with a dearth of simple displacement assays for primary ligand screening. Thus, use of small molecules for in vitro and in cellulo study of G4s is constrained by two major factors: (1) relatively small number of G4-selective molecules with an optimum affinity that balances the desirable ligand versus fluoroprobe behavior, and (2) smaller number of molecules that can be used across DNA and RNA G4s. In this study, we report a dimeric cyanine small molecule, B6,5, which can be used in fluorescence displacement assays to screen G4-specific ligands for DNA and RNA G-quadruplexes. We have previously reported the ability of dimeric carbocyanine dye to selectively bind DNA G4 structures [20]. B6,5 is a benzothiazole-based dimeric dicarbocyanine dye and we examine it as an alternative to TO considering the modest selectivity of the latter for G-quadruplex over duplex DNA [21,22]. While probes like N-TASQ (N-Template-Assembled Synthetic G-quartet) [23], Pyro TASQ [24], GTFH (G-quadruplex-triggered fluorogenic hybridization probe) [25] and ThT (Thioflavin T) [26] have been successful in probing RNA G4 structures in cellular context, there is no report of small molecules with an application to screen RNA G4-specific ligands. Notably, ThT is able to probe nucleolar RNA and DNA G4 structures under in vitro and in vivo conditions [27]. However, there is no report of an in vivo screening assay for RNA G4-specific ligands. Further, ThT is limited to probing nucleolar G4 structures and does not bind cytoplasmic RNA G4s and mitochondrial DNA G4s. For these reasons, ThT cannot be considered as a universal G4 specific probe. The evaluation of G4-specific ligands in cellulo was successfully reported by Zhang and co-workers using the fluorescent probe IMT (*N*-Isopropyl-2-(4-N,Ndimethylanilino)-6-methylbenzothiazole) towards G4 ligands PDS (Pyridostatin), TmPyP4, San, and RHPS3 [28]. The IMT-based detection system is limited to evaluation of DNA G-quadruplex specific compounds. By leveraging its suitable G4-binding behavior, in this work we report the ability of B6,5 to screen both DNA and RNA G4-specific ligands under in vitro and in cellulo conditions. Putative G4 forming sequences are known to be concentrated in key regulatory sites such as oncogene promoters, untranslated exonic regions, replication origins, and telomeres. The enrichment of G4 in cancer cells proposes G4 to be an attractive target in developing anticancer molecules [29]. G4 inducing or stabilizing molecules such as telomestetatin, BIBR1532, 6OTD, and CM02 have shown anticancer effects by exerting DNA damage in cancer cells or suppressing expression of genes, with putative G4 forming sequences frequently upregulated in cancers [30]. In this regard, we have used known G4-interacting ligands such as quarfloxin, TMPyP4, piperine, and olaparib. We observe that B6,5 is also able to stabilize DNA G-quadruplex structures without altering the native secondary structures of the quadruplexes. The dual property of stabilization and sensing of G4s by B6,5 is an attempt towards development of theranostic molecules [31].

Fluorescent probes for G4 DNA can alter the topology or thermal stability of G4 DNA to a varying extent [6]. The dimeric cyanine dye B6,5 stabilizes the G4 conformation in telomeric and c-myc derived sequences as seen from CD and PAGE data. This dye–G4 interaction causes increase in fluorescence intensity of B6,5. The G4-responsive fluorescence of B6,5 was leveraged for comparing in vitro and in cellulo binding behavior of DNA and RNA G-quadruplexes with different ligands.

## 2. Results and Discussion

### 2.1. Taq Polymerase Stop Assay to Establish Ligand Potential of B6,5

The *Taq* polymerase stop assay has been widely used to evaluate the role of ligands on G-quadruplex stability [6,32,33]. The assay relies on ability of G4 structures to interfere with the movement and performance of DNA polymerases such as *Taq*. Disruptions caused by G4 topologies manifests into truncated products that are effectively visualized by gel electrophoresis. To test the ligand-like potential of B6,5 (structure in Figure 1), we have used several G-rich oligonucleotides that are derived from telomeric and oncogene promoter sequences which are listed in Table 1. 3G, 4G, and H50 are telomeric derived sequences and C49 is c-myc oncogene promoter derived sequence. We performed *Taq* polymerase stop assays on all listed DNA sequences in presence of variable amounts of B6,5. Considering that we use unlabeled oligonucleotides, visualization of PAGE gel was performed using SYBR Gold staining. Quadruplex formation in all the sequences (listed in Table 1) used for this study was verified using CD Spectroscopy (Appendix A). CD spectroscopy is an accepted technique for in vitro identification and characterization of G-quadruplexes. The inherent chirality in G-quadruplexes manifests into distinctive bands in three spectral regions, 235–245 nm, 264–270 nm, and 280–297 nm. Specific G4 topologies are identified according to the combination of CD bands observed in these spectral regions. Thus, parallel G4 topology is commonly associated with CD maxima at 265 nm and minima at 245 nm, while antiparallel is associated with CD maxima at 295 nm and minima at 260 nm. In addition, hybrid G4 topologies are usually interpreted from presence of CD maxima at 295 nm and 260 nm with CD minima at 245 nm. The decrease in intensity of full-length products obtained after polymerase activity represent G4 stabilization by B6,5. As shown in Figure 2 and Figure 3, B6,5 is a G4-stabilizing ligand for both intermolecular (3G) and intramolecular (4G) quadruplex structures, respectively. The stabilizing effect of B6,5 is also observed in case of C49 and H50 (see Appendix A). A greater ligand-dependent decrease of full-length product intensity is observed for the sequences 4G and H50. This corresponds to a greater affinity and stabilizing effect of B6,5 on the corresponding hybrid G4 structures as compared to the ones formed by 3G and C49.

### 2.2. In Vitro Dye Displacement Assay for DNA and RNA G4 Structures Using B6,5

We have recently reported the distinctive self-assembly and G4 recognition by dimeric cyanine dyes [20]. The aggregates of these dyes are fluorescence-quenched in aqueous media and exhibit significant fluorescence enhancement upon interaction with G4 DNA. B6,5 is a G4-selective “turn on” fluorescent probe that forms the basis of our dye displacement assay. This dimeric dicarbocyanine dye shows strong propensity to form fluorescence quenched aggregates [21]. B6,5 displays similar G4-binding and fluorescence behavior compared to the dimeric carbocyanine dye that we have previously reported (see Appendix A). Preliminary screening of G4-specific molecules by use of dye displacement assay is attractive considering the challenges associated with duplex/quadruplex selectivity. In these assays, addition of a G4-selective ligand to a solution containing fluorescent probe and quadruplex DNA results in probe displacement and concomitant reduction in fluorescence intensity of fluoroprobe. The change in fluorescence intensity is indicative of the binding affinity of ligand for G4 DNA/RNA. However, this is subjective to binding modes of competing ligands and we understand that if binding mode of a ligand is different from the dye, the results cannot be accurately assessed. The working concentrations of ligand, fluorescent probe, and DNA play a crucial role in such dye displacement assays. The TO-based assay has been used for comparing the behavior of several known G4-interacting ligands such as TMPyP4, quarfloxin, piperine, and olaparib [34]. Interestingly, while TMPyP4 [35], piperine [36], and quarfloxin [37,38] are well-known G4 DNA stabilizers, there are only a few indirect reports on the interaction of olaparib with quadruplex DNA [39]. While the fluorescence quantum yield of TO in conjunction with G4 DNA is impressive, known G4-stabilizing ligands do not display remarkable displacement of the fluorophore in the assay. We compare the performance of B6,5 with TO as part of this work. TO is a suitable control for B6,5 in the present experiments based on its established end-stacking mode of binding to G4 structures that is also followed by the dimeric cyanine dyes [20,40]. To this end, we calculated the percentage fluorescence decrease of B6,5 and compared it with the same of thiazole orange. As mentioned previously, a greater drop in fluorescence intensity of the reporter dye used in the assay correlates with a stronger binding ligand. As shown in Figure 3B and Figure 4A, the fluorescent probe B6,5 reveals the variation in affinity of several G4-selective ligands. These experiments were performed with DNA G4 formed by 3G and 4G. We extended the application of B6,5 to assess RNA G4-selective ligands and Figure 4 conveys the displacement of B6,5 from RNA G4 structure upon interaction with RNA G4 selective ligand. This is an attempt towards in vitro screening of RNA G-quadruplex specific ligands. The increase in percentage displacement of B6,5 with increase in ligand concentration is because of greater ligand –G4 complexation and simultaneous expulsion of B6,5. While quarfloxin and TMPyP4 have greater affinity for intermolecular G4 structure 3G as compared to piperine and olaparib, the intramolecular G4 structure 4G displays high affinity for all the ligands tested.

From our results, TMPyP4, quarfloxin, and olaparib display very high affinity for RNA G4 structures resulting in over 60% displacement of B6,5 upon addition of only 1 molar equivalent of ligands. Notably, piperine does not display selective RNA G4 binding (Figure 4C). RNA G4 is thermodynamically more stable and less polymorphic than DNA G4. Thus, the behaviour of RNA G4 stabilizing molecules may not mirror the behaviour of DNA G4 ligands. The primary screening of G4-targeting molecules is performed only to test their G4 affinity. TMPyP4 is known to unfold very stable RNA G4 structures which suggests its affinity for RNA G4. The poor selectivity of first generation G4 ligands such as TMPyP4 between G4 and duplex structures must also be considered as a primary limitation. We used the TO displacement assay (Appendix AA,B) as a reference assay in the context of DNA G4 structures. Comparison of B6,5 behavior with TO in these assays reveal the superior performance of B6,5 in capturing engagement with G4-interacting ligands. It should be noted in this context that the binding behavior of G4-selective ligands is strongly influenced by quadruplex topology and molecularity. A probe deployed in G4 ligand screening assay should balance the selectivity for G4 structures with suitable binding affinity, that facilitates displacement by a wide range of ligands. Further, the dye displacement assay is most applicable for G4 ligands that do not overlap the emission spectrum of B6,5 in the 650–800 nm range. Our results show that the displacement assay of B6,5 captures the influence of G4 topology on ligand binding in a more nuanced manner as compared to TO. The results obtained with B6,5 are consistent with reported affinities of the ligands [36,41]. Notably, there are no direct reports of olaparib being a G4-stabilizing ligand. Thus, our current observations pertinent to G4-binding by olaparib would benefit from deeper scrutiny and are being pursued separately.

### 2.3. Binding Stoichiometry of B6,5 with DNA and RNA G-Quadruplexes

We examined the binding stoichiometry of B6,5 with DNA G-quadruplex 4G and RNA G-quadruplex RG4 using fluorescence-derived Jobs Plots. The total molar concentration of B6,5 and RG4 were held constant in these experiments while their mole fractions were varied. As shown in Figure 4B and Figure 5, the inflection points for the Jobs Plots of 4G and RG4 with B6,5 were observed at 0.6 and 0.4, respectively. These correspond to binding stoichiometry for 4G:B6,5 as 2:3 and for RG4:B6,5 as 2:1.

### 2.4. Interaction of B6,5 with RNA G-Quadruplexes

The affinity and selectivity of dimeric carbocyanine molecule B6,5 for RNA G-quadruplex RG4 was evaluated by CD spectroscopy (Appendix A) and fluorescence spectroscopy (Appendix A). CD has been widely used to characterize RNA G-quadruplexes [42,43,44]. The signifiers of parallel G4 topology, namely positive band around 260 nm and negative band at 240 nm, are clearly observed for RG4. While it is true that G-quadruplexes have a positive peak at 210 nm, this greatly depends on the buffer conditions used. Notably, a recent report of G-quadruplex in LINP1 lncRNA was characterized using CD did not display a 210 nm peak albeit folding into parallel G-quadruplex [45]. B6,5 does not alter the secondary G-quadruplex structure adopted by RNA (Appendix A). However, RNA G4 causes disaggregation of B6,5 leading to increase in fluorescence intensity with increase in molar equivalence of B6,5 (Appendix A). The CD spectra in Appendix A suggests that ligand binding does not alter topology of RNA G4. The subtle changes at 200–220 nm in the CD spectra correspond to minor G4-stabilization upon ligand binding. To further validate that disaggregation of B6,5 is caused by interaction with G4 structures of RNA, we performed atomic force microscopy (AFM) of B6,5 alone and B6,5 in the presence of RNA G4. Figure 6 clearly demonstrates the disaggregation of B6,5 in presence of RNA G-quadruplex. The behavior of B6,5 towards RNA G-quadruplex in conjunction with its behavior toward DNA G-quadruplexes indicates its capability to interact with both type nucleic acids. Such a probe can be developed as a useful tool for screening G4 ligands and imaging G-quadruplexes in cells.

### 2.5. Visualization of Cellular G-Quadruplexes in Fixed Cells Using B6,5

The in vitro interaction of B6,5 probe with DNA and RNA G4 structures led us to investigate its behavior in cellulo. Antibodies such as BG4 and 1H6 have enabled visualization of DNA and RNA G-quadruplexes in fixed cells. However, antibodies have unique constraints surrounding stability, immunogenicity and cell permeability leading to difficulties in live cell imaging. Small molecules are promising alternatives to these immune-based assays and probes such as N-TASQ, ThT, and IMT have displayed attractive potential for visualizing cellular RNA and DNA G4 structures. As shown in Figure 7, fixed cell confocal imaging of HeLa cells stained with B6,5 and co-stained with DAPI (4′,6-diamidino-2-phenylindole) reveals localization of B6,5 in nucleoli and in cytoplasmic regions close to nucleus that possibly correspond to rough endoplasmic reticulum. For better clarity the HeLa cells stained with B6,5 without DAPI overlay are shown in Figure 7F, which shows nucleoli foci in red. Nucleoli staining by B6,5 is prominent in live cell imaging (Figure 8A) in contrast to fixed cell imaging. The off-target effects of B6,5 cannot be neglected considering that the plethora of macromolecules present in cells can cause B6,5 disaggregation. Nevertheless, affinity of binding of the dimeric dye B6,5 is significantly greater with G4 in contrast to proteins. Furthermore, the brightly fluorescent regions observed (Figure 8A) correspond to regions that are known to harbor stable G-quadruplexes [27]. B6,5 staining possibly reveals rDNA and rRNA G-quadruplexes in nucleoli. We performed DNase and RNase digestion to understand the targets of dye staining. As shown in Figure 6C and Figure 7B, B6,5 is able to target both DNA and RNA structures. Urea denatures all nucleic acid secondary structures including G4 and thereby reducing the potential targets of B6,5 and a concomitant decrease in fluorescence of the dye (Figure 7D). The regain of fluorescence in Figure 7E upon rinsing of urea with PBS (pH 7.4) suggests recovery of RNA/DNA secondary structures [27].

### 2.6. In Cellulo Dye Displacement Using B6,5

We investigated the in cellulo behavior of B6,5 with respect to displacement by quadruplex-selective ligands. This was pursued by the treatment of cells with G4-specific ligands followed by fluorescence microscopic observation of B6,5 staining activity. The analogous in vitro displacement assay showed variable decreases in fluorescence intensity, which should be ideally captured in cells by CLSM(Confocal laser scanning microscope). An intracellular dye displacement assay is attractive for screening cell permeability and quadruplex-selectivity of ligands within the complex cellular environment at the same time. Accordingly, we treated B6,5 stained HeLa cells with the ligands olaparib, piperine, and quarfloxin. A clear reduction in fluorescence intensity of B6,5 is observed upon the ligand treatments, as shown in Figure 9A–D. Interestingly, olaparib treatment manifests in the greatest drop in B6,5 fluorescence followed by quarfloxin and piperine. This suggests that olaparib is destined for a larger number of cellular targets including G4 structures. Quarfloxin is an established G4-binding ligand which results in B6,5 displacement from quadruplex structures and a corresponding drop in fluorescence intensity. We used ThT staining as a reference (Appendix A) partly based on the attractive cell permeability, G4-binding and reported in cellulo fluorescence quenching upon treatment with the G4-ligand pyridostatin. The current in cellulo screening is limited to compounds that do not interfere with the emission profile of B6,5. Thus, while TMPyP4 is a strong G4 stabilizing ligand, it interferes with emission readout of B6,5, thereby constraining the format. Nevertheless, an in cellulo screening tool for G4-selective ligands provides a more holistic perspective on the effect of G4-specific ligands within cells. For example, it is possible that a modestly G4-stabilizing ligand exhibits excellent cell permeability and attractive selectivity thereby improving its potential use. The in vivo screening is therefore capable of providing cost and time-effective G4-binding leads.

### 2.7. Live Cell Imaging of HeLa Cells Using B6,5 and Its Displacement Using Quarfloxin

To validate that B6,5 fluorescence originated from interactions with quadruplex structures we performed a displacement assay in real-time using 1 molar equivalent of quarfloxin. Quarfloxin was chosen because it is an established G4 stabilizing ligand, and also, because it does not interfere with the emission wavelength of B6,5. To evaluate the uptake of B6,5 and time taken for its displacement by quarfloxin, we performed a time course fluorescence measurement in HeLa cells using CLSM. We can distinctly observe nucleoli stained by B6,5 in Figure 8A and upon addition of quarfloxin the fluorescence intensity getting reduced suggesting displacement of B6,5. As shown in Figure 8A, B6,5 distinctively stained nucleoli and cytoplasmic regions in HeLa cells. We attempted to co-stain the nucleus of live HeLa cells with Hoechst 33,342, but presence of B6,5 appears to restrict its penetration inside the nucleus. The nucleoli foci of B6,5 were more clearly observed in live cells in contrast to fixed cells. Quarfloxin is an effective G4 stabilizing ligand and due to superior affinity for G4 structures it was expected to displace B6,5. The displacement of B6,5 is evident from the corresponding fluorescence quenching over a period of time as shown in Figure 8A–F. Furthermore, the drop in intensity between Figure 9A–C suggests that the displacement of B6,5 by quarfloxin is fulfilled within a narrow time window and is clearly recorded by live cell imaging. Quick displacement of B6,5 represents strong affinity of quarfloxin for cellular G4 structures. The fluorescence quenching is instantaneous using quarfloxin within less than 4 min demonstrating in cellulo application of dye displacement assay.

## 3. Material and Methods

### 3.1. Materials

Solvents and chemicals used for experiments are HPLC grade. Methanol (Finar), potassium chloride (SRL), sodium hydroxide pellets (SRL), acrylamide solution (40%)—(Invitrogen, Waltham, MA, USA), APS (Invitrogen, Waltham, MA, USA), TEMED (Sigma, Asao Ku, Kawasaki-shi, Kanagawa, Japan), 10× TAE (Himedia, Swastik Disha Business Park, L.B.S. Marg, Mumbai, India), Trizma base (Sigma, Asao Ku, Kawasaki-shi, Kanagawa, Japan), ethylenediaminetetraacetic acid (Finar, Gujarat, India), SYBR Gold (Invitrogen, Waltham, MA, USA), and 10 bp DNA ladder (Promega, Madison, WI, USA) were used. Bis 6,5 dimeric carbocyanine ligand was synthesized based on a reported procedure and provided as a gift by Dr. Prathap Reddy Patlolla [21].

### 3.2. Oligonucleotides

All oligonucleotides were procured from IDT in lyophilized form. The stock solutions of 100 µM were made in nuclease free water and were kept at −20 °C for long-term storage. Table 1 in results and discussion displays oligonucleotides that were used for in vitro experiments. All sequences when taken out of −20 °C and were heated at 95 °C before use to break non-specific secondary structures, if any.

### 3.3. Polymerase Stop Assay

Polymerase stop assays were performed based on reported procedures with modest modifications necessary for analysis by non-radioactive methods. Appropriate amounts of template DNA sequences (100 nM, unless specified otherwise) were mixed with respective primers (50 nM, unless specified otherwise) and were allowed to anneal in annealing buffer (50 mM Tris pH 8.0, 10 mM MgCl_2_) by heating at 95 °C for 5 min and 650 rpm, followed by gradual cooling to room temperature. Furthermore, the annealed primer–template was mixed with 1X polymerase extension buffer (40 mM Tris HCl pH 8.0, 1 mM MgCl_2_, 5 mM DTT, 100 ug/mL BSA, 250 µM ATP and 0.1% NP40), 100 mM KCl, 100 µM dNTPs and ligand B6,5 at suitable concentrations to make a final volume 20 µL. This was followed by 60 min of incubation at 37 °C and 650 rpm to allow interaction of B6,5 with quadruplex structures. Finally, primer extension reaction was initiated by adding *Taq* polymerase followed by incubation of 60 min, 650 rpm at 50 °C for c-myc and 40 °C for telomeric DNA (3G, 4G and H50). The reaction was stopped by an adding equal volume of stop buffer (95% formamide, 20 mM EDTA, 0.05% bromophenol blue and 0.05% xylene cyanol). The samples were analyzed on 15% denaturing PAGE containing 100 mM KCl in gel and running buffer. The gel was visualized by post-staining with 1× SYBR Gold (Invitrogen, Waltham, MA, USA). The intensity of full-length product bands observed in gel were quantified using FIJI software (previously known as ImageJ). Bands were selected using rectangle tool and blank was selected of same dimension from a region of gel where no bands appeared either above or below the full-length product. The blank was marked as lane 1 and bands were marked as lane 2. From analyse-gels-plot lanes we obtained a graph indicating intensities. Furthermore, the peak ends of the graph were joined using the wand tool and each area was clicked in a synchronized manner starting from top to bottom. Data of lane 1 was subtracted from data of lane 2 and the derived data was plotted as a column graph.

### 3.4. Fluorescence Dye Displacement Assay

Template DNA of 0.5 µM concentrations was mixed with 100 mM KCl and 10 mM Tris-HCl pH 7.2 and was heated at 95 °C for 5 min and cooled down on ice for 1 h. Dye (B6,5 or TO) was mixed with this solution at a concentration of 1 µM and was incubated for 60 min at 37 °C. Ligands like TMPyP4, piperine, quarfloxin, and olaparib were added in specified equivalent concentration to DNA and solution was mixed well with pipetting. All samples were then allowed to equilibrate for 5 min at room temperature before fluorescence intensity was captured using Horiba fluorimeter. The excitation and emission maxima of B6,5 are 640 nm and 666 nm, respectively, while the spectral scan was done from 650–800 nm. Similarly, the excitation and emission maxima of TO were 501 nm and 533 nm, respectively, while the spectral scan was done from 510–750 nm. The percentage of dye displacement was calculated as follows:% of dye displacement = 100 − [(fluorescence intensity of sample)/fluorescence intensity of standard) × 100](1)

The fluorescence intensities of sample and standard were blank corrected average fluorescence intensities from three samples. The standard fluorescence intensity was calculated for DNA + TO or DNA + B6,5 at their respective fluorescence emission maxima.

### 3.5. Jobs Plot

We used fluorescence spectroscopy for deriving Job’s plot that can give an insight on binding stoichiometry of B6,5 with DNA or RNA G-quadruplex structures. In this method, the total molar concentration of DNA/RNA and B6,5 was kept constant at 5 µM, but their mole fractions were varied. Pre-folded DNA or RNA G-quadruplex was used in the experiment in the presence of 100 mM KCl and 1×PBS (pH 7.4) buffer and was allowed to equilibrate at 37 °C for 1 h. The fluorescence spectra were captured using JASCO fluorimeter at excitation wavelength 640 nm and emission maxima 666 nm. Samples of DNA or RNA at 0.2, 0.4, 0.6, 0.8, and 1.0 mole fractions were used for the calculation of binding stoichiometry. The formula used for calculating mole fraction was as follows:X_n_ = X_n_/(X_n_ + X_d_)(2)
where X_n_ = mole fraction of DNA/RNA (nucleic acid). X_d_ = mole fraction of dye B6,5.

### 3.6. Fixed Cell Imaging

HeLa (cervical carcinoma) cells were cultured in high glucose Dulbecco’s modified Eagle’s medium (DMEM) containing 10% fetal bovine serum (FBS) for 24 h. For fixing, cells were treated with 4% paraformaldehyde for 20 min. The permeabilization of cellular stains such as DAPI, propidium iodide, ThT (Thioflavin T), or B6,5 was achieved by treating fixed cells with 0.1% Triton X-100. For enzyme treatment, cells were treated with 0.1 mg/mL DNase and 1 mg/mL RNase for 1 h, followed by co-staining with DAPI (10 µM) and B6,5 (1 µM) for 15 min. The cells were visualized under a confocal laser scanning microscope (CLSM) equipped with an oil immersion 63× objective. CLSM images of ThT, propidium iodide, and DAPI were collected under excitation wavelength of 405 nm, 561 nm, and 405 nm, respectively. Further 633 nm laser was used for excitation of B6,5. For urea treatment, HeLa cells were incubated with 7 M urea solution for 15 min to achieve denaturation and then for urea rinsed, the cells were washed with PBS (pH 7.4) buffer solution. The cells were stained with ThT (5 µM) or B6,5 (1 µM) and nuclei were co-stained with DAPI or propidium iodide. For dye displacement assays, HeLa cells were treated with DAPI and B6,5 (1 µM) for 15 min, followed by treatment with 1 molar equivalent of ligands (quarfloxin, piperine, and olaparib) for 20 min before acquiring CLSM images. The in cellulo dye displacement by ligands was calculated using Fiji software. The boundary of each cell was marked in Fiji software and the background was subtracted from it. Then, using ROI manager Integrated density (IntDen), per unit area of cell was measured for three cells in an image. The values are plotted as an average of IntDen per unit area from three different images for each sample.

### 3.7. Live Cell Imaging

For live cell imaging, HeLa cells cultured in DMEM with 10% FBS were treated with 1 µM B6,5 and Hoechst 33,342 (1 µg/mL) for 1 h, followed by treatment of 1 molar equivalent of quarfloxin. CLSM images were acquired at 63× magnification at an interval of every 2 min for 10 min duration. The excitation wavelengths for B6,5 and Hoechst 33,342 were 633 nm and 405 nm, respectively.

### 3.8. Circular Dichroism Spectroscopy Studies

CD experiments were performed on a JASCO J-815 spectropolarimeter. A quartz cuvette with 4 mm path length was used for recording spectra, from 200 to 320 nm at 1 nm bandwidth with a response time of 1 s. The scanning speed was 50 nm min^−1^ and the reported data are average of three scans, at room temperature. CD analysis consisted of two parts: (1) using KCl as stabilizing monovalent ion and (2) titration against ligand B6,5 at a fixed DNA concentration. The oligomer concentration was fixed at 4 µM in 10 mM Tris-HCl buffer of pH 7.2 and 100 mM KCl. After addition of KCl, the template was incubated at 37 °C for 60 min before recording the spectra.

CD titration was performed at a fixed oligonucleotide concentration (4 μM) and increasing molar equivalents of ligand B6,5 from 1 equivalent to 5 equivalents in 10 mM Tris-HCl buffer of pH 7.2 with 100 mM KCl. After each addition of the compound, the reaction was stirred and allowed to equilibrate for at least 60 min at 37 °C after which CD spectrum was recorded. A background CD spectrum of the corresponding buffer was subtracted from the average scan for each sample. All final analysis of the data was carried out using Origin 8.0 (OriginLab Corp., Northampton MA, USA).

### 3.9. Atomic Force Microscopy

We used clean, sterile microscope glass slide carefully cut in 1 × 1 cm^2^ for Atomic Force Microscopy (AFM) analysis and immediately prior to DNA deposition, and the top layer of the slide was cleaved using Scotch tape to reveal an atomically flat surface. B6,5 was diluted to a working concentration of 1.0 µM and a drop was carefully placed on the slide. In the sample slide, pre-formed RNA G-quadruplex was diluted to a concentration of 1.0 µM in buffer (100 mM KCl, 10 mM Tris–HCl, pH 7.2). Droplets (20 μL) were deposited onto freshly cleaved glass slide. Excess dye and RNA was rinsed off with MilliQ water (Millipore System, Burlington, MA, USA) and the water was wicked from the surface using tissue paper. The glass slides were allowed to dry overnight in desiccator at room temperature. Imaging was performed using multimode 8 (Bruker, Billerica, MA, USA) atomic force microscope in tapping mode. AFM tip (Scanasyst) with a nominal spring constant of 0.4 Nm^−1^ and 60–90 kHz was employed for the experiments. Analysis was done using Gwyddion software.

## 4. Conclusions

This study is an attempt towards understanding and evaluating compounds that can act as a probe and ligand for quadruplex structures which may have plausible theranostic applications. In this regard, we have reported that small molecule B6,5 is related to DNA G4 stabilization and can also be employed to screen G4 specific ligands. While the affinity of B6,5 for DNA G4 structures was already expected, here, we reported its selectivity for RNA G4 structures using CD, AFM, and fluorescence spectroscopy. The small molecule had characteristics to recognize universal G4 structures and was seen to be a suitable alternative to Thiazole orange for primary screening of G4 ligands by displacement assay. We extended the in vitro displacement assay to cellular displacement assay for primary screening of G4 specific ligands, because it can be rapid and cost efficient. The cell permeability and specificity of B6,5 for G4 structures was validated and displacement of B6,5 by quarfloxin was observed in real-time in live cells. This work is an attempt towards developing a molecule with attractive quadruplex-specific theranostic properties.

## Figures and Tables

**Figure 1 molecules-26-01400-f001:**
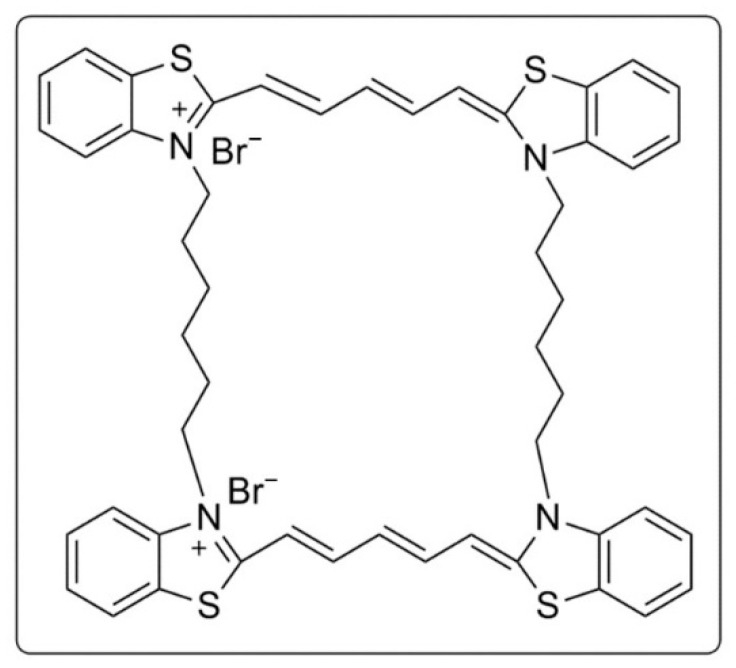
Structure of dimeric carbocyanine dye B6,5.

**Figure 2 molecules-26-01400-f002:**
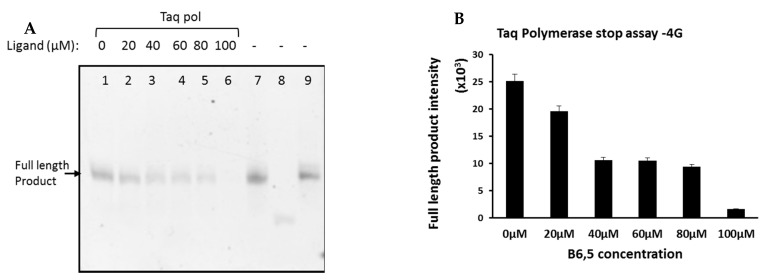
(**A**) *Taq* polymerase stop assay of 4G DNA sequence analyzed using 15% denaturing PAGE in presence of 0 to 100 µM of ligand B6,5. (Lane: 1–6) The ligand concentration and polymerase addition is as indicated in the gel image. The band in lane 7 represents template-primer in buffer without polymerase and ligand as a control. The bands in lanes 8 and 9 represent primer and template respectively. The incubation time for *Taq* polymerase was 1 h at 40 °C. (**B**) On the right hand side is the graphical representation of band intensity (lane 1–6) analyzed using ImageJ software. The error bar indicates the group error obtained on analyzing three consecutive gels having same sample set.

**Figure 3 molecules-26-01400-f003:**
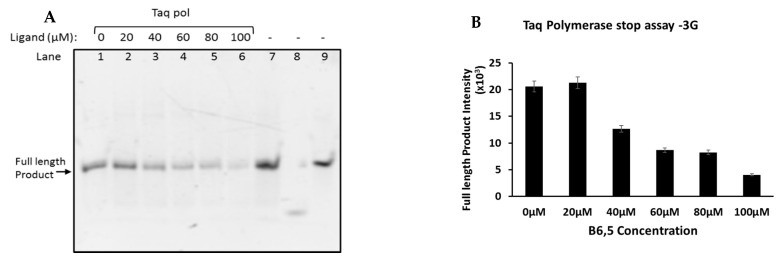
(**A**) *Taq* Polymerase stop assay of 3G DNA sequence analyzed using 15% denaturing PAGE in presence of 0 to 100 µM of ligand B6,5. (Lane: 1–6) The ligand concentration and polymerase addition is as indicated in the gel image. The band in lane 7 represents template-primer in buffer without polymerase and ligand as a control. The bands in lanes 8 and 9 represent primer and template respectively. The incubation time for *Taq* polymerase was 1 h at 40 °C. (**B**) On the right hand side is the graphical representation of band intensity (lane 1–6) analyzed using ImageJ software. The error bar indicates the group error obtained on analyzing three consecutive gels having same sample set.

**Figure 4 molecules-26-01400-f004:**
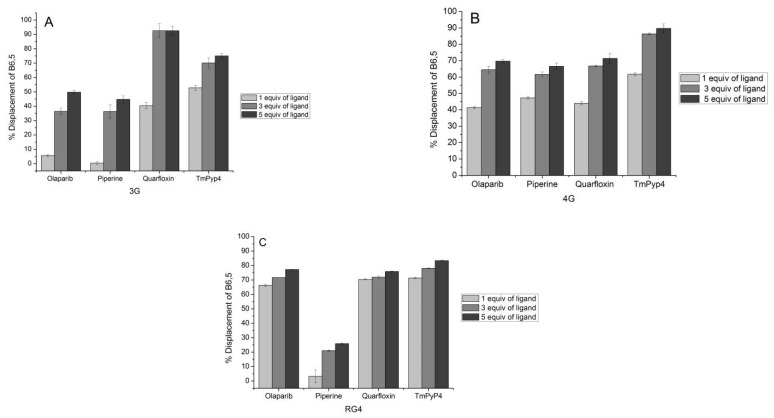
Affinity of ligands olaparib, piperine, quarfloxin, and TmPyP4 analyzed using B6,5 displacement assay in DNA G-quadruplexes (**A**) 3G and (**B**) 4G and also in (**C**) RNA G-quadruplex.

**Figure 5 molecules-26-01400-f005:**
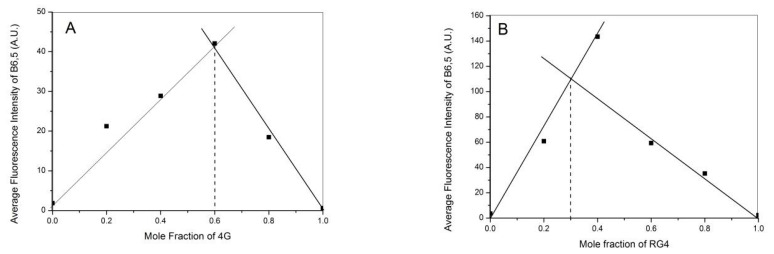
Job’s Plot for binding stoichiometry of B6,5 with quadruplexes (**A**) 4G and (**B**) RG4.

**Figure 6 molecules-26-01400-f006:**
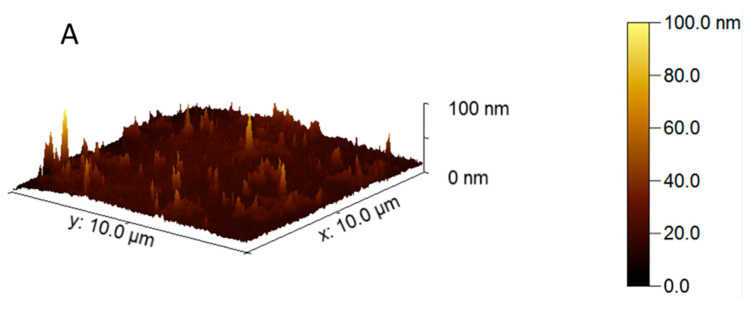
Atomic force microscopy of (**A**) B6,5 and (**B**) B6,5 + RG4 demonstrating disaggregation of B6,5 dye by RNA G-quadruplex.

**Figure 7 molecules-26-01400-f007:**
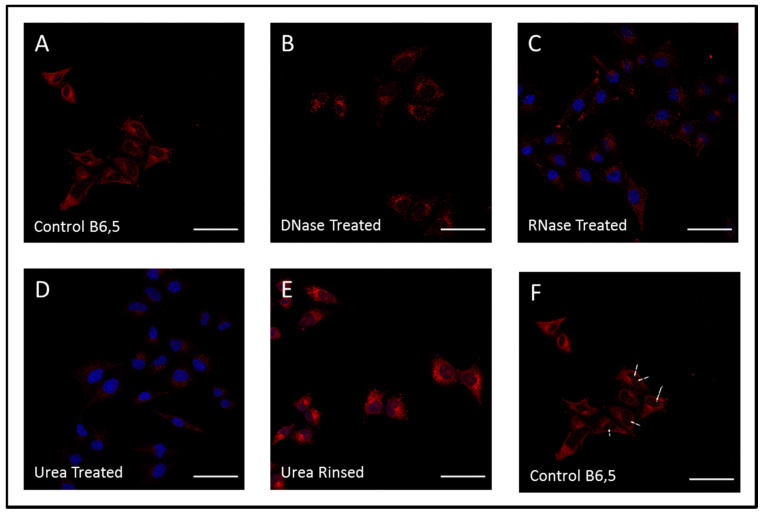
Confocal laser scanning microscope (CSLM) imaging of fixed HeLa cells with (**A**) B6,5, (**B**) DNase, (**C**) RNase, (**D**) Urea treated, (**E**) Urea rinsed with PBS (pH 7.4), and (**F**) B6,5 stained cells without DAPI channel overlay. The nucleus is co-stained using DAPI. Scale bar 50 µm. For clarity images are presented in pseudo colors of red for B6,5 and blue for DAPI.

**Figure 8 molecules-26-01400-f008:**
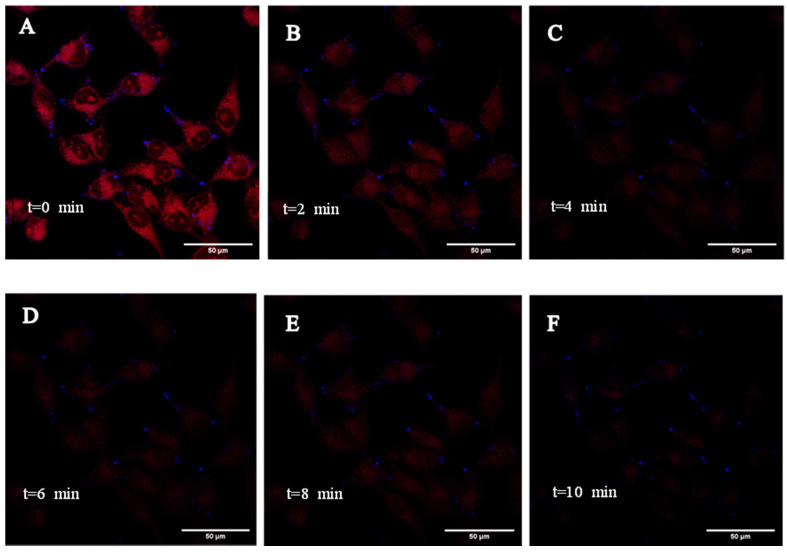
Time lapse live cell imaging of HeLa cells stained with B6,5 (1 µM) and co-stained with Hoechst 33,342 and treated with 1 molar equivalent of quarfloxin. The images were captured at fixed z-value over a period of 10 min at every 2-min interval (**A**–**F**). For clarity, images are presented in pseudo colors of red for B6,5 and blue for Hoechst 33342.

**Figure 9 molecules-26-01400-f009:**
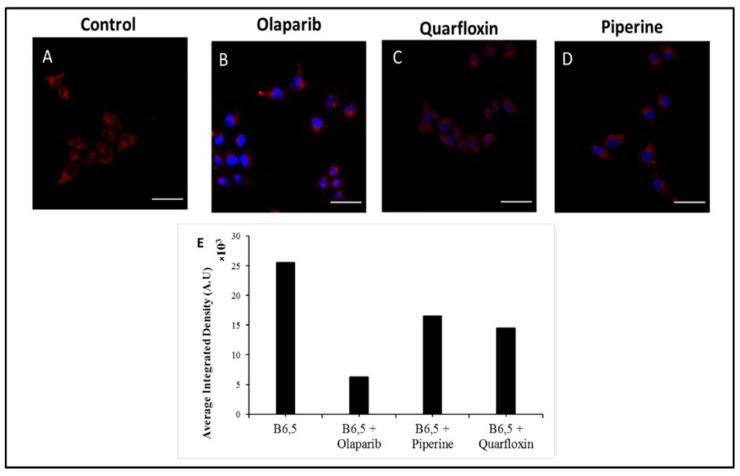
CLSM images of B6,5 stained fixed HeLa cells (**A**) control; without ligand treatment, (**B**) Olaprib treated, (**C**) Quarfloxin treated, and (**D**) Piperine treated cells, which shows fluorescence quenching of B6,5. For clarity, images are presented in pseudo colors of red for B6,5 and blue for DAPI. The relative fluorescence quantification of B6,5 and ligand treated cells is graphically represented in (**E**).

**Table 1 molecules-26-01400-t001:** Oligonucleotide sequences used in the current study. Primer binding region is given in italics and G-runs are underlined in each template sequence.

Sr. No.	Template Name	Sequence (5′ to 3′)
1	3G	TTAGGGTTAGGGTTAGGGTTA*CCTAGGCATCCTCCAGTTCCTGGAGTCAGTG*
2	4G	AGGGTTAGGGTTAGGGTTAGGGTTA*CCTAGGCATCCTCCAGTTCCTGGAGTCAGTG*
3	P31 (Primer 3G/4G)	CACTGACTCCAGGAACTGGAGGATGCCTAGG
4	H50	AGGGTTAGGGTTAGGGTTAGGGGCCAC*CGCAATTGCTATAGTGAGTCGT*
5	c49	TGAGGGTGGGGAGGGTGGGGAAGCCAC*CGCAATTGCTATAGTGAGTCGT*
6	Primer H50/c49	ACGACTCACTATAGCAATTGCG
7	RG4	AAGGAAGGGGAAGUCAGGUGGGGCCUGGGGAACCAGGAAGCGGGGAACAGG

## Data Availability

Data is contained within the article or supplementary information.

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
