# Peer review of "Assessing G4-Binding Ligands In Vitro and in Cellulo Using Dimeric Carbocyanine Dye Displacement Assay"

_molecules, 2021, doi:10.3390/molecules26051400_

Round 1

Reviewer 1 Report

This paper fits the scope of the journal. The manuscript provides an innovative and efficient method to visualize G4 structure and screening different G4-ligands by displacement assay.
The manuscript appears well-written and composed in a clear fashion style. In general, the procedures employed seem rational.
However, there are several issues with the manuscript, which lead me to conclude that this work is not suitable for publication in Molecules, MDPI, in its current form and some changes would need to be implemented before the publication.
Major point:
The authors confirmed the G4 formation of the template sequences used by CD. It is clear that H50, C49, 3G and 4G DNA oligonucleotide sequences are folded into G4 structures from CD spectra in Figure S2A (in any case, I suggest adding a short and explanatory comment on the CD results, to make it understandable even to those who do not work on the G4 field, see below).
Otherwise, I am not pretty sure that RG4 RNA oligonucleotide is G4-folded.
It is known that RNA-G4s have a parallel topology and, usually, the difference between G4 parallel topology and an A-form is the band around 210 nm (positive for G4 and negative for A-form).
How do the authors are sure RG4 is G4-structured in that conditions?
From Figure S7, it seems that the addition of increasing amounts of B6,5 ligand induce some changes in the RG4 structure (following at 200-220 nm), but they are not comparable with the DNA situation (Figure S2A).
Are there other studies in the literature that confirm the G4-folding of the sequence reported by the authors as RG4? Have the authors the opportunity to demonstrate the G4-folding for RG4 with another technique?
If not, I suggest to try to repeat the experiments with a more known RNA-G4 like TERRA-RNA. Or to remove the studies related to this particular sequence.
Minor points:
1. In Line 28 the authors defined the word “G-quadruplex” with “G4 structure”, reported in brackets. Nevertheless, they subsequently use the “G4” abbreviation to identify the word “G-quadruplex” (as in lines 38, 39, 46, 77 and so on). I suggest to change the definition of “G-quadruplex” in line 28 only with the more appropriate acronym “G4” (removing the word “structure”).
2. Line 32. The words "in vitro" must be written in cursive, to maintain the format of the manuscript. See line 416 for the same reason.
3. Please add appropriate references for the sentences:
- “While probes like N-TASQ, Pyro TASQ, GTFH and ThT have been successful in probing RNA G4 structures in cellular context” (Line 62)
- “ThT is able to probe nucleolar RNA and DNA G4 structures under in vitro and in vivo conditions” (Line 65)
- “ThT is limited to probing nucleolar G4 structures and does not bind cytoplasmic RNA G4s and mitochondrial DNA G4s” (Line 67)
- Lines 69-72. The authors referred to the work of Zhang et al. on the fluorescent probe IMT. Please add the correspondent reference properly.
4. Table 1. The authors say they have carried out experiments on the c-MYC sequence reported in Table 1, but none of the reported sequences are named “c-MYC” (as in line 318). Please check the correct template names.
Moreover, the caption reported that “G-runs are underlined in each template sequence” but no underlines are present for RG4. Please check.
The tables must have the caption above the table and not below. Please reverse the order. Correct the word “current” (double R) and “underlined” (an N is missing). Finally, I suggest to the authors to move Table 1 after its proper citation in the main text (e.g. after the dot in line 95).
5. Lines 94-95. The authors reported that they studied “G-rich oligonucleotides that are derived from telomeric and oncogene promoter sequences”. I suggest reporting in this sentence the template name of the G4 sequences into brackets so that they are appropriately identified as “telomeric” (3G and 4G) and “oncogene promoter” sequences.
6. Lines 95-96. The authors say that they “performed Taq polymerase stop assays on 3G and 4G sequences in presence of variable amounts of B6,5”. Nevertheless, they also performed Taq polymerase stop assay on C49 and H50 (results reported in Figure S3A and S3B). Please, correct the entire paragraph emphasizing that the study was done on all available DNA sequences.
7. Lines 98-100. The authors confirmed the G4 formation of the template sequences used, using CD spectroscopy. Please add a proper reference and a short and exhaustive description of the results obtained with CD to make them understandable to everyone, also out from G4 field.
8. Please standardize Figure 2 in Supp. Inf.
Both the y axis reported the unit of measurement “mdeg”, but one is reported as “ellipticity” and the other as “CD”. Please make it equal. Moreover, “wavelength” is written with a lowercase letter in 2SA and with a capital letter in S2B. Please make it equal. In the caption, please add a space between “10” and “mM” to maintain the format of the manuscript.
9. The captions of Figures 1 and 2 are almost the same. I suggest joining Figures 1 and 2 as a single Figure 1A and B with a single caption. The same could be done for Figure S3 and S4.
10. Line 106. Why do the authors referred to 4G and H50 structures as “distorted parallel”? Is it no longer correct to say G4 hybrid-structures? The CD dichroic features are more like hybrid topologies.
11. Line 128. The caption of Figure S5 reported the name “Dye 1” which is not present in the text but I supposed to be the “dimeric carbocyanine dye”. I suggest to the authors to add “Dye 1” in the main text in brackets after its definition. Moreover, please correct the “u” of “2 uM” with “μ” and check the name “c-Myc” to maintain the format of the manuscript: sometimes it has a hyphen, sometimes not.
12. Lines 137-139. Please add a proper reference for “TO-based assay”, always to make this manuscript accessible even to non-G4-experts.
13. Line 141. Even if the authors report that there are only a “few indirect reports” for olaparib, it would be appropriate to add at least one proper reference.
14. Paragraph 2.4. Although the disaggregating effect that the RNA sequence has on B6,5 is evident, I am not sure that RG4 sequence is G4-folded by CD results. Please see the major point and modify the entire paragraph 2.4. consequently.
15. Line 215. Change the capital letter of the word “Fixed”, I do not think is necessary.
16. Line 219 - Line 273. Write “figure 6F” and “figure 8A” with capital letters to maintain the format of the manuscript.
17. Line 231. Please add informations on PBS buffer, like pH value. See line 363 for the same reason.
18. Line 233. In the caption of Figure 6 appear the abbreviation “CLSM” that is explicated as “confocal laser scanning microscope” only in line 359. Please, add the entire name in the main text before using the abbreviation.
19. Line 296. The brand “Invitrogen” must be in brackets to maintain the format of the manuscript. Moreover, “Sigma” has to be written with the capital letter.
20. Line 298. The authors reported that “Bis 6,5 dimeric carbocyanine ligand was synthesized based on reported procedure”, a proper reference must be added.
21. Line 303. The words "in vitro" must be written in cursive, to maintain the format of the manuscript.
22. Lines 311 and 313. Please modify “MgCl2” putting the number in the subscript (MgCl2)
23. Lines 311 and 316. Add a space between “650” and “rpm”, to maintain the format of the manuscript.
24. Line 314. Add a space between “100” and “ug/mL”, to maintain the format of the manuscript.
25. Line 320. Add a space between “20” and “mM” to maintain the format of the manuscript.
26. Line 325. Add a space between “100” and “mM” to maintain the format of the manuscript.
27. Lines 344-345. Please correct: “100mM KCl” - “pH7.4” - “37ºC” adding the proper spaces to maintain the format of the manuscript. Moreover, the time is indicated with “1 hr”, but in the rest of the main text the authors referred to “hours” only with “h”. Please uniformize. For the same reason, see also lines 353-357-375.
28. Line 377. “10 minute” must be plural.
29. Line 382. Please modify “min-1” putting “-1” as apex of “min”.
Please, check the extra spaces between the word all over the manuscript (as in line 70, 156, 244, 354, 381).
Moreover, check and correct the Abbreviated Journal Name in the references

Reviewer 2 Report

Review of the manuscript “Assessing G4-binding Ligands in Vitro and in Cellulo Using Dimeric Carbocyanine Dye Displacement Assay” by Desai N. et al.

Comments to Authors

This paper describes the development of compounds that can act as a probe and ligand for quadruplex structures for theranostic applications. The paper is generally properly composed, but in some parts chaotically written. Based on the following comments, I advise making major modifications in this paper before resubmitting.

The Reviewer would like to address the following comments:

  1. The Authors should find more appropriate references for chip-based analysis and ligand fishing (line 47) in which application of these techniques to G-quadruplex analysis will be described.
  2. The figure describing the structure of B6,5 should be included in the main body of text.
  3. The better organization of theoretical information in Introduction part, straightforward outlining the purpose of research and potential results would make the paper more transparent and available to the reader.
  4. The information of folding topology of G4 should be placed in line 96, the place where these particles are mentioned for the first time.
  5. The legend of the graph and the figure description of Supplementary Figure S2 should be unified and improved.
  6. Could the Authors explain the reason of lack of bands representing shorter products in Taq polymerase stop assay? Why the intensity of bands representing full length product and control reaction without enzyme and ligand is the same? Does the disappearance of the bands in the Authors opinion reflect the inhibition of Taq polymerase reactions?
  7. In the Reviewer opinion the results of Taq polymerase stop assay for H50 and 3G are similar. 
  8. It was also found that there is a mistake in the Supplementary Figure S6 (the figure for 3G has probably been posted twicely).
  9. The single stranded, duplex and hairpin structures should be use in vitro B6,5 displacement assay as control compounds to prove the specificity of the interaction of the dye with G4. What is more, the test analysis of the displacement of TO by B6,5 would be valuable.
  10. Why binding stoichiometry was only counted for 4G and RG4?
  11. Why DAPI is invisible in Figure 6A?
  12. Why the KCL was used in 15% denaturing gel? In the Reviewer opinion denaturing conditions are applied to have no structure in the line and addition of stabilizing cations have no sense in that case.
  13. There were some editorial mistakes: results and discussion (line 303) and Supplementart Figure in whole Supplementary file.
  14. The paper suffers from the lack of authors deeper conclusions of the reported literature data.

Reviewer 3 Report

The manuscript “Assessing G4-binding Ligands in Vitro and in Cellulo Using 1 Dimeric Carbocyanine Dye Displacement Assay” by Bhaskar Datta et al. reports on the interaction of the molecule named B6,5 with G3/G4 nucleic acid and RNA sequences. The G-quadruplexes interaction with ligands is attracting a lot of attention and studies on the matter are increasing in number, e.g. the manuscript will definitely attract readers. The employed methods include stop assay, FID, CD, confocal microscopy and AFM. The prevalent concept behind most of the methods is the registration of the appearance/disappearance of fluorescence/luminescence due to the displacement of one “dye” by another. In general, the selection and combination of methods supports the results and the conclusions and is appropriate. The manuscript readability is average and may eventually be improved by the authors as some paragraphs and parts tend to rephrase previous ones.

In my opinion the present state of the manuscript is quite good and the recommendation is that it can be accepted for publications after some minor corrections.

Minor points:

P1L28 . A Hoogsten hydrogen bonding in DNA is usually formed between a purine and pyrimidine. The G4 motif is stabilized by GGGG interactions. It is possible that in some case the stacking of the g4 produces a Hoogsten type of hydrogen bonding interaction but is it a general feature of G4? Please check and provide a reference or correct. Ref 1 is focused on oncogene promoters.

P1L32 ( and in conclusion) in vitro – please italicize (italic)

P2L81 . “../ derived sequences as seen from CD and PAGE data “ please include figure reference.

P3L86 :Table 1. Please move the table captions before the table.

P11L324 3.4. Fluorescence Dye Displacement Assay . What is the exact model of the Horiba fluorimeter used? The TO is in visible range 500-530 nm while the B6,5 is almost in the IR. Same detector?

P13L411 Conclusions “In this regard, we have reported the ligand like DNA G4 stabilization property of small molecule B6,5.” Please rephrase; unclear

Supplementary Figure S3and S4

Supplementart to Supplementary, Supplementary Figure S3and S4 .

Figure 1, S3and S4. Please provide a reference for ImageJ and what method/plugin/protocol has been used for intensity “extraction”?

This is a general remark and may be omitted but the references may be updated to include more recent data. The  present state  basically ends  2018 .

Round 2

Reviewer 2 Report

The paper was improved by the Authors and the Reviewer would like to thank you for taking his opinion under consideration. Most of the addressed issues were sufficiently implemented. However, there are still some points, which it would be worth  to improve prior publication.

1. The Authors should find more appropriate references for chip-based analysis and ligand fishing (line 47) in which application of these techniques to G-quadruplex analysis will be described.
We have now provided more suitable references for chip-based analysis and ligand fishing as suggested. We thank the reviewer for suggesting as such.

In Reviewer opinion Busto N, Calvo P, Santolaya J, Leal JM, Guédin A, Barone G, Torroba T, Mergny JL, García B. Fishing for G-Quadruplexes in Solution with a Perylene Diimide Derivative Labeled with Biotins. Chemistry. 2018 Aug 6;24(44):11292-11296. doi: 10.1002/chem.201802365. Epub 2018 Jul 10. PMID: 29797628 is more adequate reference for ligand fishing. 

2. The better organization of theoretical information in the Introduction part, straightforward outlining the purpose of research and potential results would make the paper more transparent and available to the reader.
We thank the reviewer for the insight. We have made changes in text in the introduction/background at lines 49, 52, 59, 62, 66, 77 and 97 that would hopefully make the paper easier to follow.

In Reviewer opinion the Introduction part should be further improved. For example, sentences: “One   approach   towards conception of such probes is to associate fluorophores with exhibit G4-selective ligands in a  modular  manner” and  “Another  approach  involves fusing molecules that stabilize G-quadruplexes with a fluorescent agent” seems to have the same meaning. The information are chaotically spread along the text. The goal of the research is not enough drawn.  

3. The information of folding topology of G4 should be placed in line 96, the place where these particles are mentioned for the first time.
We have made the change as suggested.

In the Reviewer opinion folding topology is still not enough stressed. Some conclusions can be made based on Supplementary Figure S1.

4. Could the Authors explain the reason of lack of bands representing shorter products in Taq polymerase stop assay? Why the intensity of bands representing full length product and control reaction without enzyme and ligand is the same? Does the disappearance of the bands in the Authors opinion reflect the inhibition of Taq polymerase reactions? The shorter bands in polymerase stop assay are easier to observe when DNA is radiolabelled. In this work, we have used the safer and most cost-effective alternative of post-staining of gels by use of SYBR Gold. The detection limit of SYBR Gold is 25pg and it is likely that the shorter DNA bands are formed at lesser quantities than the detection limit thereby making them invisible in the gel. We observed other reports of PCR stop assay of G-quadruplex when visualized by SYBR Gold staining also representing one single band of product (Huang, Wei-Chun, et al. "Direct evidence of mitochondrial G-quadruplex DNA by using fluorescent anti-cancer agents." Nucleic acids research 43.21 (2015): 10102-10113.). The intensity of bands of full-length product and control reaction without enzyme and ligand is same because the template concentration is same across all samples.

In the Reviewer opinion the progress of extension reaction should result in increase of band intensity. The extended strand could bind more SYBR GOLD. Based on the presented data it could be only supposed that ligand force G-quadruplex formation, what resulted in disappearance of band due to lack of SYBR GOLD penetration. Could the Authors state their opinion on the above?  

5. Why binding stoichiometry was only counted for 4G and RG4?
The binding stoichiometry of all listed DNA oligonucleotides was same and therefore only 4G is represented.

This information should be mentioned in the text.

6. There were some editorial mistakes: results and discussion (line 303) and Supplementary Figure in whole Supplementary file.
We have made the necessary changes.

Please write Supplementary Figure (…) instead of Suplementary Figure (…).